# The Antidepressant Paroxetine Reduces the Cardiac Sodium Current

**DOI:** 10.3390/ijms24031904

**Published:** 2023-01-18

**Authors:** Ingmar S. Plijter, Arie O. Verkerk, Ronald Wilders

**Affiliations:** 1Department of Medical Biology, Amsterdam Cardiovascular Sciences, Amsterdam UMC, University of Amsterdam, 1105 AZ Amsterdam, The Netherlands; 2Department of Experimental Cardiology, Heart Center, Amsterdam Cardiovascular Sciences, Amsterdam UMC, University of Amsterdam, 1105 AZ Amsterdam, The Netherlands

**Keywords:** antidepressant drugs, Na_V_1.5 channels, sodium current, action potential, cellular electrophysiology, patch clamp recordings, HEK-293 cells, cardiomyocytes, computer simulations

## Abstract

A considerable amount of literature has been published on antidepressants and cardiac ion channel dysfunction. The antidepressant paroxetine has been associated with Brugada syndrome and long QT syndrome, albeit on the basis of conflicting findings. The cardiac voltage-gated sodium channel (Na_V_1.5) is related to both of these syndromes, suggesting that paroxetine may have an effect on this channel. In the present study, we therefore carried out patch clamp experiments to examine the effect of paroxetine on human Na_V_1.5 channels stably expressed in human embryonic kidney 293 (HEK-293) cells as well as on action potentials of isolated rabbit left ventricular cardiomyocytes. Additionally, computer simulations were conducted to test the functional effects of the experimentally observed paroxetine-induced changes in the Na_V_1.5 current. We found that paroxetine led to a decrease in peak Na_V_1.5 current in a concentration-dependent manner with an IC_50_ of 6.8 ± 1.1 µM. In addition, paroxetine caused a significant hyperpolarizing shift in the steady-state inactivation of the Na_V_1.5 current as well as a significant increase in its rate of inactivation. Paroxetine (3 µM) affected the action potential of the left ventricular cardiomyocytes, significantly decreasing its maximum upstroke velocity and amplitude, both of which are mainly regulated by the Na_V_1.5 current. Our computer simulations demonstrated that paroxetine substantially reduces the fast sodium current of human left ventricular cardiomyocytes, thereby slowing conduction and reducing excitability in strands of cells, in particular if conduction and excitability are already inhibited by a loss-of-function mutation in the Na_V_1.5 encoding *SCN5A* gene. In conclusion, paroxetine acts as an inhibitor of Na_V_1.5 channels, which may enhance the effects of loss-of-function mutations in *SCN5A*.

## 1. Introduction

Pharmaceutical drugs play a crucial role in today’s society. Life expectancy is increasing, new diseases are emerging, and new improved drugs are being developed and marketed. This is reflected by the increasing trend in drug prescriptions [1]. The rising incidence of drug development is a double-edged sword: on the one hand, a broader spectrum of drugs is available, but on the other hand, these new drugs can be associated with as yet unknown side effects [2]. The safety profile is assessed during drug development, but the safety of a drug can never be fully guaranteed because it is tested in a highly selected patient population [3]. This can lead to a drug being marketed and a potentially dangerous side effect not being noticed until several years later [2].

A potentially dangerous side effect of a drug is the occurrence of a cardiac arrhythmia [4,5], i.e., a condition in which the heart cannot regulate its rhythm correctly. A disturbed cardiac rhythm or conduction pattern can give rise to various symptoms, including sudden cardiac arrest [6,7]. It is now well established from a variety of studies that certain drugs can cause arrhythmias [8]. Even among commonly used drugs to treat a specific non-cardiac disease, differences in risk of out-of-hospital cardiac arrest (OHCA) may exist. To illustrate, in a recent study on the association between various antidepressants and OHCA, it was found that several antidepressants can increase the risk of OHCA [9].

An either congenital or acquired abnormality in the cardiac voltage-gated sodium channel (Na_V_1.5), which is responsible for the rapid upstroke of the myocardial action potential (AP), can cause a potentially life-threatening cardiac disorder, e.g., long QT syndrome (LQTS) or Brugada syndrome (BrS) [10,11]. Although the antidepressant paroxetine, a selective serotonin reuptake inhibitor (SSRI), was not overrepresented in OHCA cases using antidepressants [9], it has been associated with both BrS and LQTS. In a case report on a syncopal episode of a patient taking a daily dose of 10 mg paroxetine, Sawhney et al. [12] showed that paroxetine withdrawal led to complete resolution of the clinically observed BrS type 1 ECG pattern. Of note, the daily dose of 10 mg is a rather low dose given the regular dosage range of 20–50 mg once daily [13], up to doses of 62.5 and 80 mg/day reported by Dunner et al. [14] and Reis et al. [15]. This holds even more so because of the disproportionate and highly nonlinear increase in plasma drug levels upon increasing doses of paroxetine [16].

Data on the association of paroxetine with LQTS are somewhat conflicting. Edwards et al. [17] studied the electrocardiographic effects of a four week treatment with a regular daily dose of paroxetine (30 mg) in 20 patients (11 on paroxetine and 9 on placebo). Apart from a small but significant increase in QRS width (see below), they did not observe any changes in electrocardiographic parameters, including the QTc interval. From a systematic search on the risk of QT prolongation among SSRIs, Funk and Bostwick [18] concluded that the use of paroxetine per se is not sufficient to induce a clinically significant QT prolongation, although it may induce or enhance QT prolongation in combination with other drugs. In particular, Lim et al. [19] found that paroxetine enhanced the QT prolongation induced by the class Ic antiarrhythmic agent flecainide, which is a potent inhibitor of the cardiac sodium current [20] and has therefore been used as a provocative challenge to unmask BrS and thereby identify patients at risk [21]. Erfurth et al. [22] reported serious QT prolongation in response to paroxetine in two patients with a rather complex medical history. Furthermore, maternal use of paroxetine has been associated with QTc interval prolongation in exposed neonates [23,24]. Paroxetine has also been implicated in cases of torsade de pointes, but always in the setting of additional risk factors [25].

Both BrS and LQTS, more specifically LQTS type 3, are associated with loss-of-function mutations in the *SCN5A* gene, which encodes the pore-forming α-subunit of the Na_V_1.5 channel [10,11]. Considering the association of paroxetine with both BrS and LQTS, an effect of paroxetine on cardiac sodium channels is highly likely. This hypothesis seems supported by findings of Yokota et al. [26] and Edwards et al. [17]. Yokota et al. [26] recorded ECGs from anesthetized mongrel dogs and found a statistically significant 25% increase in QRS width, which is importantly modulated by the cardiac sodium current [27], in response to a high dose of paroxetine (10 mg/kg). In their aforementioned study, Edwards et al. [17] found a small increase in QRS width in the paroxetine-treated group that was significantly different from the small decrease that occurred in the placebo group. In line with these observations, paroxetine is classified both as a “drug preferably avoided by BrS patients” on the BrugadaDrugs.org website [28] initiated by Postema et al. [29] and as a “drug to be avoided by congenital long QT patients” as well as a “drug with conditional risk of torsade de pointes” on the CredibleMeds QTdrugs List [30] initiated by Woosley and colleagues [31].

Furthermore, it has been found that paroxetine blocks the Na_V_1.4, Na_V_1.7, and Na_V_1.8 neuronal sodium channel isoforms [32,33,34]. Because these neuronal isoforms share great homology with the cardiac sodium channel isoform, a reduced Na_V_1.5 current in response to paroxetine is plausible. These findings raise the question of how paroxetine affects the cardiac sodium channel, as no previous research has been conducted on the effects of paroxetine on the amplitude or kinetics of the cardiac sodium current. Hence, in the present study, we examined how paroxetine affects the amplitude and kinetics of the Na_V_1.5 sodium current. To this end, we carried out patch clamp experiments on human embryonic kidney 293 (HEK-293) cells stably expressing human Na_V_1.5 channels. Furthermore, we determined the effects of paroxetine on the AP of freshly isolated rabbit left ventricular cardiomyocytes in perforated patch clamp experiments. Additionally, we carried out computer simulations to assess the functional effects of the experimentally observed paroxetine-induced changes in the amplitude and kinetics of the Na_V_1.5 current on human left ventricular cardiomyocytes.

## 2. Results

### 2.1. Effects of Paroxetine on Na_V_1.5 Current in HEK-293 Cells

#### 2.1.1. Effects of Paroxetine on Peak Na_V_1.5 Current

First, we analyzed how paroxetine affects the current-voltage relationship of the peak Na_V_1.5 current in HEK-293 cells stably expressing human Na_V_1.5 channels. Figure 1a shows typical whole-cell current traces of a single HEK-293 cell upon voltage clamp steps from a holding potential of −120 mV to test potentials ranging from −80 to +35 mV (inset) in the absence (left) or presence (right) of 10 μM paroxetine, demonstrating that 10 μM paroxetine strongly inhibits peak Na_V_1.5 current. Figure 1b shows the average current-voltage relationship of the peak Na_V_1.5 current in the four HEK-293 cells to which we administered paroxetine at a concentration of 3 μM. The peak Na_V_1.5 current reached its maximum amplitude near −45 mV and was significantly decreased by paroxetine over a wide voltage range (two-way repeated measures ANOVA), with a highly similar amount of block across this voltage range. At −45 mV, the decrease in peak Na_V_1.5 current amplitude amounted to 25.4 ± 3.7% (mean ± SEM, *n* = 4). The reversal potential in the absence and presence of 3 µM paroxetine did not differ significantly (−2.0 ± 1.8 mV (control) vs. −4.6 ± 2.4 mV (3 µM paroxetine); *p* > 0.05, paired *t*-test). Thus, we exclude a reduction in driving force as mechanism for the smaller Na_V_1.5 currents in the presence of paroxetine. At paroxetine concentrations of 1 and 10 μM, this decrease was 13.9 ± 3.9% (*n* = 4) and 62.6 ± 4.5% (*n* = 3), respectively (Figure 1b, inset), again with a highly similar amount of block over a wide voltage range and again without a significant difference in reversal potential (not shown in Figure 1b). Figure 1c shows the associated dose–response curve, which was obtained by fitting the data to the Hill equation y = 1/{1 + ([paroxetine]/IC_50_)^nH^}, where y is the normalized peak Na_V_1.5 current at −45 mV, IC_50_ is the half-maximal inhibitory paroxetine concentration, and n_H_ is the Hill coefficient. The thus obtained IC_50_ and n_H_ amounted to 6.8 ± 1.1 µM and 1.14 ± 0.22, respectively (R^2^ = 0.98). The degree of inhibition was dose dependent (*p* < 0.001, one-way ANOVA).

#### 2.1.2. Effects of Paroxetine on Steady-State Activation

To determine the voltage dependence of activation of the Na_V_1.5 current, the current-voltage relationships of each individual HEK-293 cell under control conditions and in the presence of 3 μM paroxetine were corrected for driving force, normalized to maximum peak current, and fitted to a Boltzmann curve, characterized by its half-activation voltage (V_½_) and its slope factor (k). Fitting the average activation data in the absence or presence of paroxetine yielded Boltzmann curves that were virtually overlapping (Figure 2a), indicating unaltered activation. This was corroborated by the V_½_ (top) and k (bottom) values of Figure 2b (middle pair of bars), as obtained by fitting the activation data of the four individual cells to a Boltzmann curve; neither V_½_ nor k differed significantly from their paired controls (paired *t*-tests). Similar results were obtained in experiments with higher and lower concentrations of paroxetine (Figure 2b, rightmost and leftmost pair of bars, respectively).

#### 2.1.3. Effects of Paroxetine on Steady-State Inactivation

The voltage dependence of Na_V_1.5 current inactivation is important for the functional availability of sodium channels during cardiac APs [35]. In a separate series of experiments, we therefore also determined the steady-state inactivation curve (or availability curve) of Na_V_1.5 currents under control conditions and in the presence of paroxetine. Figure 2c shows the voltage dependence of steady-state inactivation under control conditions and upon administration of 3 µM paroxetine (open circles and filled squares, respectively; *n* = 5). Paroxetine induced a clear hyperpolarizing shift in steady-state inactivation, supported by a significant negative shift in the half-inactivation voltage V_½_ (Figure 2d, top, middle pair of bars), as obtained by fitting the inactivation data of each individual cell studied to a Boltzmann curve. The shift at a paroxetine concentration of 3 μM amounted to −7.3 ± 1.5 mV (*n* = 5). The −12.4 ± 2.8 mV (*n* = 3) hyperpolarizing shift that we observed at a paroxetine concentration of 10 μM was also significant, but the −3.3 ± 2.3 mV (*n* = 5) shift at 1 μM was not. These shifts in V_½_ were not accompanied by significant changes in the slope factor (Figure 2d, bottom). The apparent dose dependence of the shift in V_½_ was not statistically significant (*p* = 0.051, one-way ANOVA).

#### 2.1.4. Effects of Paroxetine on Rates of Activation and Inactivation

Changes in the rates of activation and inactivation of Na_V_1.5 current may also affect the sodium current function during cardiac APs. We therefore studied our recorded voltage clamp traces in more detail, normalizing the Na_V_1.5 current traces in order to directly assess kinetics. As illustrated in Figure 3a, we observed that the activation of the Na_V_1.5 current was not substantially affected by 3 µM paroxetine. However, its rate of inactivation was higher in the presence of the drug, resulting in Na_V_1.5 currents of shorter duration. This was quantified in detail for all used concentrations by determining the time point of half inactivation (t_50_), relative to the time point of peak current amplitude, upon a voltage clamp step from −120 to −20 mV under control conditions and in the presence of paroxetine. At concentrations of 3 and 10 µM, paroxetine significantly shortened t_50_, from 0.77 ± 0.06 to 0.58 ± 0.04 ms (*n* = 8) at 3 µM, and from 0.63 ± 0.07 to 0.36 ± 0.04 ms (*n* = 6) at 10 µM (Figure 3b, middle and right pair of bars, respectively). The inactivation rate was further quantified by determining the rate of inactivation at t_50_. This rate increased significantly at paroxetine concentrations of 3 and 10 µM, from 0.64 ± 0.05 to 0.81 ± 0.08 ms^−1^ at 3 µM, and from 0.80 ± 0.09 to 1.49 ± 0.18 ms^−1^ at 10 µM (Figure 3c). The increase in rate of inactivation turned out to be concentration-dependent, this rate approximately being doubled at a paroxetine concentration of 10 µM (Figure 3d). Repeating our analysis with voltage clamp steps to a test potential of −50 mV instead of −20 mV revealed highly similar changes in the rate of inactivation in response to paroxetine (not shown in Figure 3). This increase in rate of inactivation may, at least to some extent, underlie the observed paroxetine-induced reduction in peak Na_V_1.5 current (Figure 1b).

### 2.2. Effects of Paroxetine on Action Potentials of Rabbit Left Ventricular Cardiomyocytes

In a final set of patch clamp experiments on isolated rabbit left ventricular cardiomyocytes (*n* = 6), we assessed the effect of 3 μM paroxetine on APs in a cardiomyocyte background under close-to-physiological conditions. The concentration of 3 μM was chosen because it gave rise to a considerable amount of changes in Na_V_1.5 current in voltage clamp experiments on HEK-293 cells (Figure 1, Figure 2 and Figure 3) without too extreme effects that might prevent excitability of cardiomyocytes completely. Figure 4a shows typical AP recordings at a pacing frequency of 1 Hz under control conditions and in the presence of paroxetine (gray and black lines, respectively). Paroxetine significantly lowered the maximum AP upstroke velocity of the six cardiomyocytes (Figure 4a, inset; Figure 4b; *p* < 0.05 vs. control, paired *t*-test) and also significantly reduced the AP amplitude (Figure 4a,c; *p* < 0.05 vs. control, paired *t*-test). Maximum upstroke velocity was decreased by 8.7 ± 2.5%, and because the upstroke is largely driven by Na_V_1.5 channels [35], this indicates that the Na_V_1.5 current is also reduced in a cardiomyocyte environment at close-to-physiological conditions. No significant differences were found in other AP parameters, including resting membrane potential, AP plateau potential, and AP duration at 20, 50, and 90% of repolarization (Figure 4d–f).

Many Na_V_1.5 current inhibitors—including, for example, methylflavonolamine and lidocaine [36], hesperetin [37], eleclazine [38], carbamazepine [39], mexiletine [40], and aripiprazole [41]—generate a shift in steady-state inactivation to more negative potentials (as shown for paroxetine in Figure 2c,d) that is accompanied by a slowing of recovery from inactivation. Under close-to-physiological conditions, i.e., at a potential of −85 mV and 37 °C [35], recovery from inactivation of the sodium current in cardiomyocytes is not fully complete within 1 s, which explains why the maximum AP upstroke velocity at a pacing frequency of 3 Hz is smaller than at 1 Hz under control conditions (Figure 5a, gray lines; Figure 5b, gray bars). In the presence of paroxetine, this frequency-induced decrease in maximum AP upstroke velocity is enhanced, as demonstrated in Figure 5a (gray and black lines) and Figure 5b (gray and black bars), which show data from 5 of the 6 cardiomyocytes that were used for Figure 4. For the sixth cardiomyocyte, we did not record APs at 3 Hz in the presence of paroxetine. Under control conditions, an increase in pacing frequency from 1 to 3 Hz results in a 16.0 ± 4.5% decrease in maximum AP upstroke velocity, but in the presence of 3 µM paroxetine this decrease amounts to 33.1 ± 2.6% (Figure 5c; *p* < 0.01, paired *t*-test). This more pronounced decrease in maximum AP upstroke velocity in the presence of 3 µM paroxetine strongly suggests that paroxetine does not only inhibit Na_V_1.5 current through a negative shift in steady-state inactivation (Figure 2) and an increased rate of inactivation (Figure 3), but also through a slower recovery from inactivation.

### 2.3. In Silico Experiments

#### 2.3.1. Effects of Paroxetine on Peak Na_V_1.5 Current

We observed that paroxetine affects the inactivation of Na_V_1.5 current in HEK-293 cells through a negative shift in the half-inactivation voltage (Figure 2) and an increase in the rate of inactivation (Figure 3). To test to which extent these changes can explain the also observed decrease in peak Na_V_1.5 current (Figure 1), we carried out in silico voltage clamp experiments, using the human left ventricular cell model by Ten Tusscher et al. [42], as updated by Ten Tusscher and Panfilov [43]. We applied the voltage clamp protocol of Figure 1a to the model cell and determined the peak amplitude of its fast sodium current under control conditions and with the changes in half-inactivation voltage and inactivation rate that we had found in our HEK-293 cells. As illustrated in Figure 6 (gray bars), the thus obtained paroxetine-induced decrease in peak Na_V_1.5 current was found to be largely due to the increase in inactivation rate, with the net effect amounting to 2.1% at 1 µM, 8.3% at 3 µM, and 25.8% at 10 µM, respectively. This is substantially less than the experimentally observed paroxetine-induced decrease in peak Na_V_1.5 current of 13.9 ± 3.9%, 25.4 ± 3.7%, and 62.6 ± 4.5%, respectively (Figure 6, black bars), demonstrating that this decrease is not solely attributable to the associated changes in Na_V_1.5 current inactivation. For example, at 3 µM, a further 18.7% reduction in peak Na_V_1.5 current of the model cell is required to arrive at the experimentally observed decrease of 25.4%.

#### 2.3.2. Effects of Paroxetine on Single Human Left Ventricular Cardiomyocytes

Next, we carried out in silico experiments to test the effects of the paroxetine-induced changes in Na_V_1.5 current characteristics on APs of single human left ventricular cardiomyocytes, using the aforementioned human left ventricular cell model by Ten Tusscher et al. [42,43]. To allow a direct comparison with our experimentally observed effects on rabbit left ventricular cardiomyocytes (Figure 4), we based our parameter settings on the Na_V_1.5 current changes observed in HEK-293 cells at a paroxetine concentration of 3 µM. Accordingly, we used a −7.3 mV shift in the half-inactivation voltage and a 26.8% increase in the inactivation rate of the fast sodium current (I_Na_) to simulate the effects of paroxetine on the model cell. Furthermore, the fully activated I_Na_ conductance was reduced by 18.7% to account for the amount of decrease in peak Na_V_1.5 current that is not attributable to the changes in inactivation, as set out in Section 2.3.1.

We assessed paroxetine-induced changes in APs elicited at 1 Hz with stimuli of 3 ms duration, as in our experiments on rabbit left ventricular cardiomyocytes. A severe loss-of-function mutation in *SCN5A* was simulated by a 50% reduction in the density of fast sodium channels, as if the heterozygously expressed mutant Na_V_1.5 channels were completely non-functional. As illustrated by the train of APs of Figure 7a (top), the overall AP shape is neither strongly affected by the simulated loss-of-function mutation nor by the simulated administration of paroxetine, apart from a marked reduction in AP overshoot. However, a closer look at the AP upstroke phase reveals that the upstroke is considerably slowed down by the mutation and upon the administration of paroxetine (Figure 7a, bottom). Maximum AP upstroke velocity (V_max_) of the wild-type cardiomyocyte is reduced by 60% (from 361 to 145 V/s) upon administration of paroxetine, whereas the already reduced V_max_ of the Na_V_1.5 mutant cardiomyocyte is further reduced by as much as 65% (from 208 to 73 V/s) (Figure 7b).

#### 2.3.3. Effects of Paroxetine on Conduction Velocity and Excitability

In a final series of in silico experiments, we tested the effects of paroxetine on conduction velocity and excitability, using one-dimensional strands of ventricular cells that are each individually described by the aforementioned human left ventricular cell model by Ten Tusscher et al. [42,43] and electrically coupled through a gap junctional conductance (g_j_; Figure 8a, inset). As in the single cell simulations of Section 2.3.2, we based our parameter settings on the Na_V_1.5 current changes observed in HEK-293 cells at a paroxetine concentration of 3 µM. A propagating AP was elicited by applying a 2 ms, 20–25% suprathreshold stimulus to the leftmost cell of the strand at a frequency of 1 Hz. At g_j_ values ranging from 10 nS to 30 µS, we determined the conduction velocity of the propagating AP under control conditions (‘wild type’) and in case of a simulated complete loss-of-function mutation in *SCN5A* (‘mutant’), and then repeated our simulations upon simulated administration of paroxetine. The thus obtained conduction velocity values are shown in Figure 8a. Under control conditions, administration of paroxetine reduces conduction velocity by ≈34% at g_j_ values ranging from 0.1 to 30 µS (Figure 8a, open and filled circles). In case of the mutation, where conduction velocity is already reduced by ≈21% due to the mutation per se (Figure 8a, open symbols), administration of paroxetine further reduces conduction velocity by as much as ≈42% (Figure 8a, open and filled squares), so that the overall reduction in conduction velocity due to the combined inhibiting effect of the mutation and paroxetine as a ‘second hit’ on top of the mutation is as large as ≈54% (Figure 8a, open circles and filled squares).

We also assessed the effects of paroxetine on excitability. To this end, we determined the stimulus current threshold over the same wide range of g_j_ values that we used in our simulations of Figure 8a. Under control conditions, administration of paroxetine increases the stimulus current threshold by 14–15% at g_j_ values ranging from 0.03 to 30 µS (Figure 8b, open and filled circles). The stimulus current threshold shows an increase of ≈10% as a result of the mutation per se (Figure 8b, open symbols). Administration of paroxetine as a ‘second hit’ further increases the stimulus current threshold by 12–18% (Figure 8b, open and filled squares), so that the overall increase in stimulus current threshold due to the combined effect of the mutation and the administered paroxetine is as large as 24–30% (Figure 8b, open circles and filled squares).

## 3. Discussion

A strong relationship has been reported in the literature between antidepressants and cardiac ion channel dysfunction [45]. As set out in the Introduction, the antidepressant of our study, paroxetine, has been linked to both BrS and LQTS, although data are limited and somewhat conflicting. Given the association of paroxetine with both BrS and LQTS and its established inhibiting effect on neuronal Na_V_1.4, Na_V_1.7, and Na_V_1.8 channel function [32,33,34], the main objective of this study was to investigate the effect of paroxetine on the functioning of the highly homologous cardiac Na_V_1.5 channel.

The main findings of the present study are as follows: (1) in HEK-293 cells stably expressing human Na_V_1.5 channels, paroxetine inhibits peak Na_V_1.5 current density in a concentration-dependent manner; (2) paroxetine does not affect voltage dependence of Na_V_1.5 current activation; (3) paroxetine inhibits Na_V_1.5 current by shifting its steady-state inactivation towards more hyperpolarizing potentials and increasing its rate of inactivation in a concentration-dependent manner; (4) paroxetine reduces V_max_ and amplitude of the AP of isolated rabbit left ventricular cardiomyocytes; (5) the latter reduction in V_max_ is larger at a higher pacing frequency, likely due to a slower recovery from inactivation in the presence of paroxetine; (6) paroxetine lowers conduction velocity and excitability in computer simulated strands of human left ventricular cardiomyocytes, thereby increasing the risk of reentry and unidirectional block [46] and thus increasing the susceptibility to cardiac arrhythmias. Thus, the present study demonstrates the effect of paroxetine as an inhibitor of Na_V_1.5 channels in HEK-293 cells and rabbit left ventricular cardiomyocytes. Furthermore, our computer simulations show that the inhibiting effect of paroxetine is increased in the setting of an Na_V_1.5 current that is already reduced due to a loss-of-function mutation in *SCN5A*.

Some caution should be taken when interpreting these results. When looking at the AP recordings of the rabbit left ventricular cardiomyocytes, one can see a significant decrease in V_max_ and APA, which both well reflect the functioning of the cardiac sodium channels [35,47], in the absence of changes in other AP parameters. These findings are in line with those of Rose et al. [48], who found that an earlier start of inactivation in the AP did not lead to major alterations in AP parameters, except for a reduction in V_max_. The observed reduction in V_max_ and APA suggests a block of Na_V_1.5 channels. However, this reduction is much smaller than one would expect from the data that we obtained in our voltage clamp experiments on HEK-293 cells. This apparent discrepancy is in line with the findings of Remme et al. [49], who noted that characteristics of sodium channels expressed in cardiomyocytes may differ from those of sodium channels expressed in cell expression systems, such as HEK-293 cells. What may have contributed to this discrepancy is that our experiments on HEK-293 cells were performed without β-subunits of Na_V_1.5, which are known to play a role in the voltage dependence of activation and inactivation of the Na_V_1.5 channel as well as in its rate of inactivation [50,51]. A further explanation may lie in the difference between the resting membrane potential of the cardiomyocytes (near −80 mV) and the holding potential in the voltage clamp experiments on the HEK-293 cells (−120 mV). Drugs can bind to channels depending on their specific state of (in)activation. The difference in potential could therefore have led to a difference in channel state and an associated difference in binding sites available for paroxetine. Yet, another explanation could be the difference in temperature at which measurements were performed and the presence of fluoride in the HEK-293 cell pipette solution, which could also alter the kinetics of the sodium channel and drug effects on the sodium channel [35,52,53,54]. Despite the apparent discrepancy in (amount of) Na_V_1.5 channel inhibition between HEK-293 cells and cardiomyocytes, the significant reduction in V_max_ (and APA) of the cardiomyocytes at close-to-physiological conditions indicates that paroxetine may be of risk for patients with Na_V_1.5 ion channelopathies, substantiating the classification of paroxetine as a drug that should preferably be avoided by BrS patients and by congenital long QT patients [28,30].

Our results were obtained at paroxetine concentrations of 1 µM and above. Unfortunately, it is difficult to estimate the actual concentration of paroxetine in cardiac tissue of patients who are treated with a daily dosage of this drug. In addition to pronounced interindividual variations in plasma paroxetine concentrations [55], paroxetine demonstrates highly non-linear pharmacokinetics, so that an increase in its dose can result in disproportionate and unpredictable increases in plasma levels [16]. The recommended therapeutic reference range is 20–65 ng/mL (corresponding to a concentration of 0.06–0.20 µM), with a laboratory alert level of 120 ng/mL [56], but much higher plasma or serum levels have regularly been observed, including levels near 0.6 µM [57], 0.9 µM [55], 1.1 µM [15], 1.2 µM [58], and 1.3 µM [16], all at regular daily doses, reflecting the aforementioned pronounced interindividual variations and highly non-linear pharmacokinetics. Paroxetine shows a high volume of distribution, with reported values of 8.7 L per kg of body weight [59], 3.1–28.0 L/kg [60], and 17 L/kg [61], suggesting that all cardiomyocytes are exposed. The distribution of paroxetine in the body is extensive, consistent with its lipophilic amine character, but with its mean cardiac specimen/blood ratio of 1.05 ± 0.43 (mean ± SD, *n* = 8), there is no specific accumulation in the heart [61]. However, it is difficult, if not impossible, to estimate the paroxetine concentration that is actually “sensed” by Na_V_1.5 channels in the cardiomyocyte membrane, yet the small but significant increase in QRS width observed by Edwards et al. [17] during a four week treatment with a 30 mg daily dose of paroxetine suggests that this concentration is near the micromolar range.

In the present study, a limited number of cells was used in each phase of the study. However, we have performed paired experiments, thus considerably raising the power of statistics for this small number of experiments, and we have used three different drug concentrations to test the effects of paroxetine on Na_V_1.5 current. The effects of paroxetine on Na_V_1.5 current density and gating properties were notable and consistent at multiple drug concentrations. In addition, the AP measurements in freshly isolated rabbit cardiomyocytes supported the principal findings of the HEK-293 cell measurements. We used three different drug concentrations to assess the IC_50_. While a larger number of concentrations is used in most studies, Turner and Charlton [62] demonstrated that a small number of concentrations is sufficient for the accurate determination of the IC_50_ through standard sigmoidal dose–response curves. In addition, our used concentrations are around the IC_50_, which enhances the reliability of the IC_50_ determination.

In conclusion, our study shows that paroxetine acts as an inhibitor of Na_V_1.5 channels through multiple actions on their electrophysiological characteristics. Our experiments on rabbit left ventricular cardiomyocytes show inhibitory effects that are not alarming per se, but may become so in case of loss-of-function mutations in *SCN5A*, supporting the classification of paroxetine as a drug that should preferably be avoided by BrS patients and by congenital long QT patients.

## 4. Materials and Methods

### 4.1. Cell Preparations

#### 4.1.1. HEK-293 Cell Culture

Experiments were conducted using a human embryonic kidney 293 (HEK-293) cell line stably expressing human Na_V_1.5 channels [63]. The cell line was cultured using DMEM (Gibco, Breda, The Netherlands) supplemented with 1% penicillin (Gibco), 1% streptomycin (Gibco), 10% FBS (Biowest, Nuaillé, France), 1% L-glutamine (Gibco), and Zeocin (200 µg/mL; Invitrogen, Breda, The Netherlands). Cells were subcultured in 25 mL flasks once a week and stored in a 5% CO_2_ incubator at 37 °C. Single cells were harvested at 80% confluence after 0.25% trypsin (Gibco) treatments of ≈2 min. On the day of patch clamp measurements, cells were trypsinized and stored in DMEM at room temperature until use.

#### 4.1.2. Rabbit Ventricular Cardiomyocytes

Animal procedures were performed in accordance with governmental and institutional guidelines for animal use in research and were approved by the Institutional Animal Care and Use Committee of the Amsterdam University Medical Centers, location Academic Medical Center.

Rabbit left ventricular cardiomyocytes were isolated from male New Zealand White rabbits by enzymatic dissociation as described previously [64]. Therefore, anesthesia was performed using 20 mg xylazine and 100 mg ketamine I.M., followed by intravenous administration of a bolus of 1000 IU heparin LEO. Animals were killed by an injection of pentobarbital (240 mg) and the hearts were excised and transported to the laboratory in cold (4 °C) Tyrode’s solution containing (in mM): NaCl 128, KCl 4.7, CaCl_2_ 1.45, MgCl_2_ 0.6, NaHCO_3_ 27, Na_2_HPO_4_ 0.4, and glucose 11; pH set to 7.4 by equilibration with 95% O_2_ and 5% CO_2_. Left ventricular cardiomyocytes were isolated using Langendorff perfusion at a constant pressure of 50 mm Hg and a temperature of 37 °C. Perfusion was conducted via the aorta with Tyrode’s solution for 15 min, after which it was replaced with low-calcium dissociation solution containing (in mM): NaCl 146.5, CaCl_2_ 0.01, MgCl_2_ 2.0, NaHCO_3_ 1.0, KH_2_PO_4_ 1.4, KHCO_3_ 3.3, glucose 11.0, and HEPES 16.8; pH set to 7.3 using NaOH. After 15 min, collagenase type B (0.15 mg/mL, Roche, Woerden, The Netherlands), collagenase type P (0.05 mg/mL, Roche), trypsin inhibitor (0.1 mg/mL, Roche), hyaluronidase (0.2 mg/mL, Sigma, Zwijndrecht, The Netherlands), protease XIV (Sigma), and creatine (10 mM) were added. The left ventricle was fractionated as perfusion pressure dropped from 50 to ≈2 mm Hg; the remnants were then further fragmented into single cells via a gyratory water bath shaker. Cells were stored at room temperature until use in a modified HEPES-buffered Tyrode’s solution containing (in mM): NaCl 140, KCl 5.4, CaCl_2_ 1.8, MgCl_2_ 1.0, glucose 5.5, and HEPES 5.0; pH set to 7.4 using NaOH.

### 4.2. Electrophysiology

#### 4.2.1. Data Acquisition

Small aliquots of cell suspension were put in an open recording chamber on the stage of an inverted microscope (Nikon Diaphot; Nikon Europe, Amstelveen, The Netherlands). The cells were allowed to adhere for 5 min after which superfusion with bath solution (for compositions, see Section 4.2.2 and Section 4.2.3 below) was started. Control measurements took place after ≈30 min. Thereafter, cells were perfused for 10 min with bath solution to which paroxetine at a concentration of 1, 3, or 10 μM was added, after which measurements were performed. With this 10 min incubation period, a stable effect of paroxetine was achieved. Paroxetine was dissolved in water as stock solution (2 mM) and stored at −80 °C. Freshly diluted paroxetine containing solutions were prepared daily.

Na_V_1.5 current and APs were measured with the ruptured and perforated patch clamp methodology, respectively, using an Axopatch 200B amplifier (Molecular Devices, Sunnyvale, CA, USA). Custom software (Scope, kindly provided by J.G. Zegers, MSc, and MacDAQ, kindly provided by Dr. A.C.G. van Ginneken) was used to record and analyze Na_V_1.5 current and APs. Borosilicate glass patch pipettes (GC100F-10; Harvard Apparatus, Waterbeach, UK) were pulled using a custom microelectrode puller, and had a 2–3 MΩ resistance after filling with the pipette solution (for compositions, see Section 4.2.2 and Section 4.2.3 below). Signals were filtered and digitized at 2 kHz and 40 kHz, respectively. For Na_V_1.5 current measurements, series resistance was compensated for 60–80%.

#### 4.2.2. Sodium Current Measurements

Na_V_1.5 currents were measured at room temperature in HEK-293 cells with stable Na_V_1.5 expression. We selected relatively small HEK-293 cells (with a membrane capacitance of 7.9 ± 0.7 pF (*n* = 11)) and used a bath solution with a relatively low Na^+^ concentration to obtain reliable voltage clamp. Bath solution consisted of (in mM): NaCl 20, CsCl 120, CaCl_2_ 1.8, MgCl_2_ 1.0, glucose 5.5, HEPES 5.0; pH set to 7.4 using CsOH. Pipette solution contained (in mM): CsF 110, CsCl 10, NaF 10, CaCl_2_ 1.0, MgCl_2_ 1.0, Na_2_ATP 2.0, EGTA 11, and HEPES 10; pH set to 7.2 using CsOH. Na_V_1.5 currents were activated from a holding potential of −120 mV followed by a 50 ms depolarizing pulse to test potentials ranging from −80 to +35 mV, in 5 mV steps (Figure 1a, inset). Cycle length was 5 s, which results in full recovery from inactivation at room temperature when using a holding potential of −120 mV [52]. For the voltage dependence of activation, the current-voltage relationships were corrected for driving force and normalized to maximum peak current. Voltage dependence of inactivation was measured at a test pulse (P2) to −20 mV for 50 ms, which was preceded by a 500 ms prepulse (P1) from a holding potential of −120 mV to test potentials ranging from −120 to +40 mV in 5 mV steps (Figure 2c, inset). Cycle length was 5 s. Voltage-dependent activation and inactivation data were fitted to the Boltzmann equation G/G_max_ = 1/{1 + exp [(V_½_ − V)/k]}, where G is the Na_V_1.5 conductance at test potential V, G_max_ is the maximum Na_V_1.5 conductance, V_½_ is the voltage of half-maximal (in)activation, and k is the slope factor (in mV). Na_V_1.5 conductance was determined by the equation G = I_Na_/(V − E_Na_), in which I_Na_ denotes the Na_V_1.5 current and E_Na_ is its reversal potential. The rate of Na_V_1.5 current inactivation was quantified by determining the time point of half inactivation t_50_, relative to the time point of peak current amplitude, upon a voltage clamp step from −120 to −20 mV. The paroxetine dose–response data were fitted to the Hill equation y = 1/{1 + ([paroxetine]/IC_50_)^nH^}, where y is the normalized peak Na_V_1.5 current at −45 mV, IC_50_ is the half-maximal inhibitory paroxetine concentration, and n_H_ is the Hill coefficient.

#### 4.2.3. Action Potential Measurements

APs were measured at 37 °C in single rabbit left ventricular cardiomyocytes. Cells were superfused with modified HEPES-buffered Tyrode’s solution (for composition, see Section 4.1.2 above), and patch pipettes were filled with (in mM): K-gluconate 125, KCl 20, NaCl 5, amphotericin B 0.44, and HEPES 10; pH set to 7.2 using KOH. APs were elicited at pacing frequencies of 1 and 3 Hz using 3 ms and ≈50% suprathreshold current pulses through the patch pipette. Potentials were corrected for the calculated liquid junction potential [65]. APs were characterized by AP amplitude, AP duration at 20, 50, and 90% of repolarization, resting membrane potential, maximum upstroke velocity, and AP plateau potential, defined as the membrane potential at 20 ms after the AP upstroke.

### 4.3. Computer Simulations

The functional effects of the paroxetine-induced changes in Na_V_1.5 currents were assessed by computer simulations of a single cardiomyocyte or a linear strand of cardiomyocytes, in either case using the Ten Tusscher et al. human left ventricular cell model [42], as updated by Ten Tusscher and Panfilov [43], to describe individual cells. When simulating action potential propagation in strands of cells, myoplasmic resistivity was set to 150 Ω∙cm (Figure 8a, inset) [44] and conduction velocity was computed across the middle third of the strand. Software was compiled as a 32-bit Windows application using Intel Visual Fortran Composer XE 2013 and run on an Intel Core i7 processor based workstation. For numerical integration of differential equations, we applied a simple and efficient Euler-type integration scheme with a 1 μs time step [66]. All simulations were run for a sufficiently long period to reach steady-state behavior.

### 4.4. Statistical Analysis

Statistical analysis was conducted using SigmaStat, version 3.5 (Systat Software, Inc., San Jose, CA, USA). All data are expressed as mean ± SEM. Significance levels were set at the 5% level when using a paired *t*-test, one-way ANOVA, or two-way repeated measures ANOVA, followed by pairwise comparison using a Student-Newman-Keuls test.

## Figures and Tables

**Figure 1 ijms-24-01904-f001:**
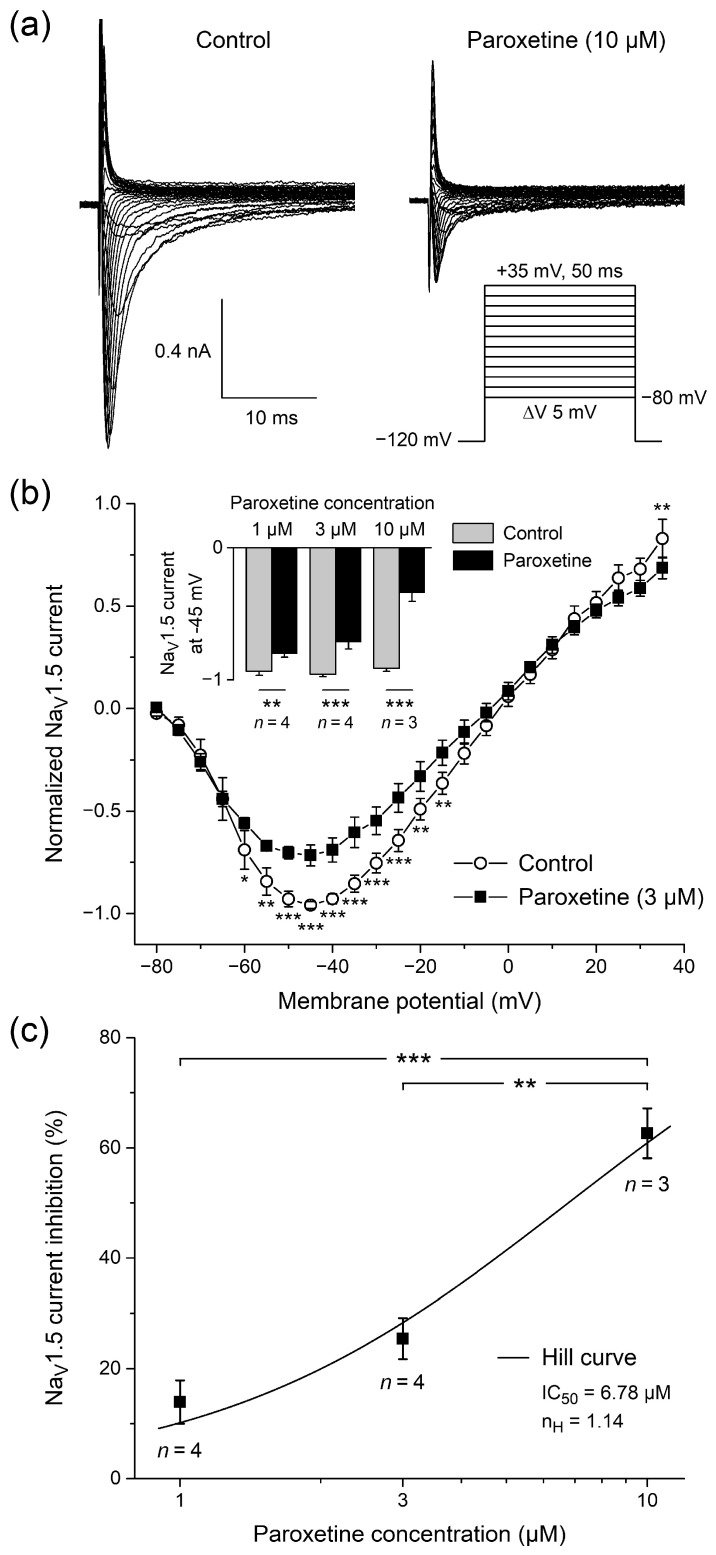
Paroxetine reduces peak current in single human embryonic kidney 293 (HEK-293) cells stably expressing human cardiac voltage-gated sodium channel (Na_V_1.5) channels. (**a**) Typical whole-cell Na_V_1.5 current traces before (left) and after administration of 10 μM paroxetine (right). Inset: voltage clamp protocol used. (**b**) Average current-voltage relationship of normalized peak Na_V_1.5 current under control conditions and upon administration of 3 μM paroxetine (open circles and filled squares, respectively; *n* = 4). Inset: normalized peak Na_V_1.5 current at −45 mV under control conditions and upon administration of 1, 3, and 10 μM paroxetine (*n* = 4, *n* = 4, and *n* = 3, respectively). * *p* < 0.05, ** *p* < 0.01, *** *p* < 0.001, two-way repeated measures ANOVA. (**c**) Dose–response curve of the inhibitory effect of paroxetine on peak Na_V_1.5 current at −45 mV. Data were fit to a Hill curve with a concentration of half-maximal inhibition (IC_50_) of 6.68 µM and a Hill coefficient (n_H_) of 1.14. ** *p* < 0.01, *** *p* < 0.001, one-way ANOVA.

**Figure 2 ijms-24-01904-f002:**
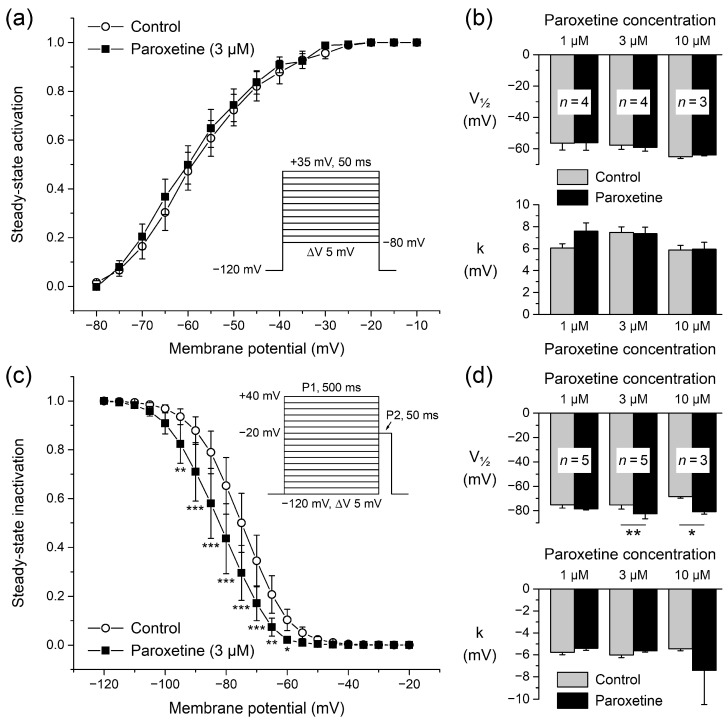
Steady-state activation and inactivation of Na_V_1.5 current in single HEK-293 cells stably expressing human Na_V_1.5 channels. (**a**) Steady-state activation curve of Na_V_1.5 current under control conditions and upon administration of 3 μM paroxetine (open circles and filled squares, respectively; *n* = 4). Inset: voltage clamp protocol used. (**b**) Average half-activation voltage (V_½_; top) and slope factor (k; bottom) values from individual Boltzmann curves to which activation data obtained under control conditions and upon administration of 1, 3, and 10 μM paroxetine (*n* = 4, *n* = 4, and *n* = 3, respectively) were fit. (**c**) Steady-state inactivation curve of Na_V_1.5 current under control conditions and upon administration of 3 μM paroxetine (open circles and filled squares, respectively; *n* = 5), as determined with a double-pulse voltage clamp protocol (inset). * *p* < 0.05, ** *p* < 0.01, *** *p* < 0.001, two-way repeated measures ANOVA. (**d**) Average half-inactivation voltage (V_½_; top) and slope factor (k; bottom) values from individual Boltzmann curves to which inactivation data obtained under control conditions and upon administration of 1, 3, and 10 μM paroxetine (*n* = 5, *n* = 5, and *n* = 3, respectively) were fit. * *p* < 0.05, ** *p* < 0.01, paired *t*-test.

**Figure 3 ijms-24-01904-f003:**
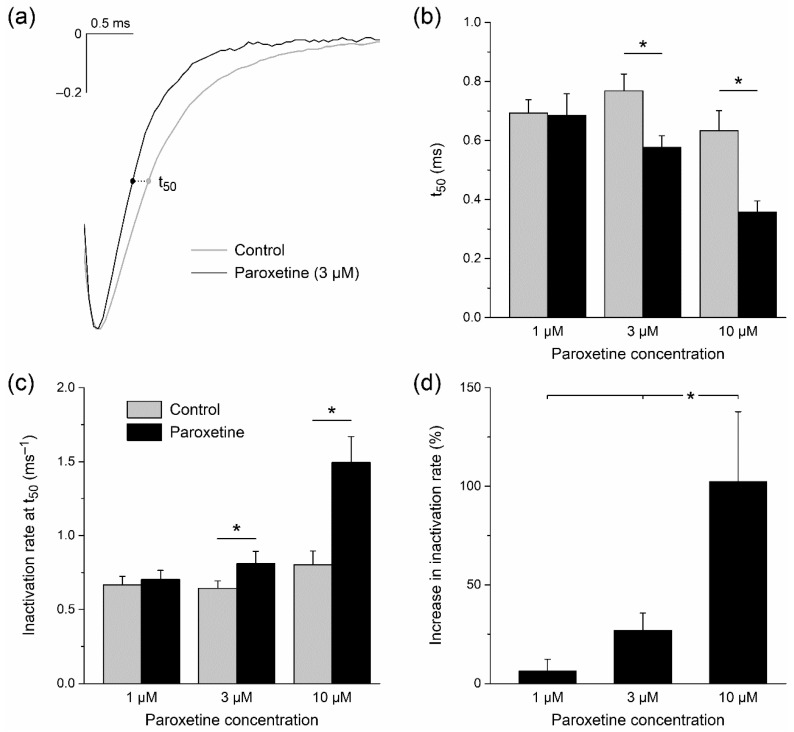
Paroxetine increases rate of inactivation of Na_V_1.5 current in single HEK-293 cells stably expressing human Na_V_1.5 channels. (**a**) Typical normalized Na_V_1.5 current traces in response to a voltage clamp step from −120 to −20 mV under control conditions and upon administration of 3 μM paroxetine (gray and black lines, respectively). The time point of 50% inactivation (t_50_) is indicated by dots. (**b**) t_50_ under control conditions and upon administration of 1, 3, and 10 μM paroxetine (*n* = 9, *n* = 8, and *n* = 6, respectively). * *p* < 0.05, paired *t*-test. (**c**) Rate of inactivation as determined by the time derivative of the signal at t_50_. * *p* < 0.05, paired *t*-test. (**d**) Percent increase in rate of inactivation at t_50_ upon administration of 1, 3, and 10 μM paroxetine (*n* = 9, *n* = 8, and *n* = 6, respectively). * *p* < 0.05, one-way ANOVA.

**Figure 4 ijms-24-01904-f004:**
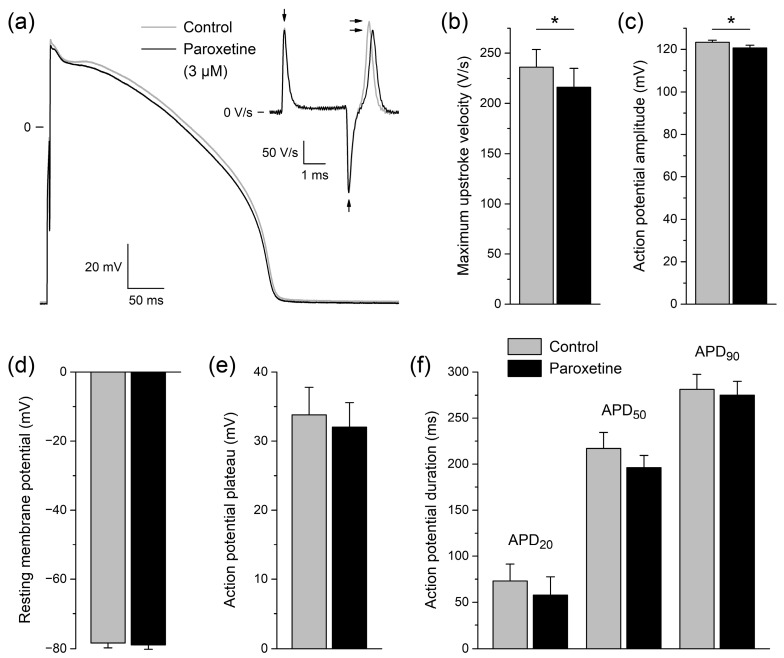
Effects of paroxetine on action potentials of isolated rabbit left ventricular cardiomyocytes. (**a**) Typical action potential (AP) of a single left ventricular cardiomyocyte under control conditions and in the presence of 3 µM paroxetine (gray and black lines, respectively). APs were elicited at a pacing frequency of 1 Hz. The upper right inset shows the time derivative of the AP signal near its upstroke. The horizontal arrows indicate the maximum upstroke velocity. The vertical arrows indicate the artefacts due to switching on and off the square stimulus of 3 ms duration. (**b**–**f**) Average AP parameters of six cardiomyocytes under control conditions and in the presence of 3 µM paroxetine (gray and black bars, respectively). (**b**) Maximum AP upstroke velocity. (**c**) AP amplitude. (**d**) Resting membrane potential. (**e**) AP plateau potential, defined as the membrane potential at 20 ms after the AP upstroke. (**f**) AP duration (APD) at 20, 50, and 90% of repolarization (APD_20_, APD_50_, and APD_90_, respectively). * *p* < 0.05, paired *t*-test.

**Figure 5 ijms-24-01904-f005:**
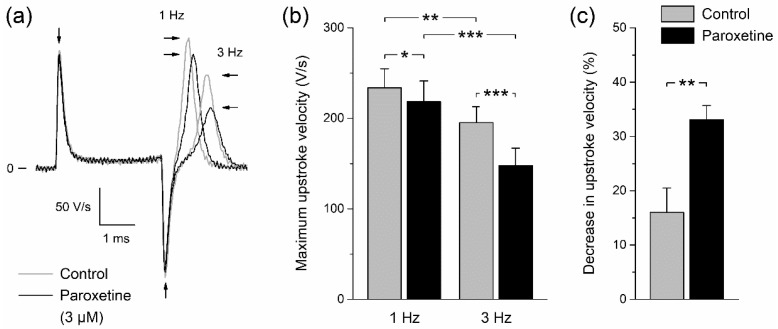
Effect of paroxetine on maximum AP upstroke velocity of isolated rabbit left ventricular cardiomyocytes is frequency dependent. (**a**) Typical time derivative of the AP signal of a single cardiomyocyte near its upstroke under control conditions and in the presence of 3 µM paroxetine (gray and black lines, respectively) at pacing frequencies of 1 and 3 Hz. The rightward and leftward horizontal arrows indicate the maximum upstroke velocity at 1 and 3 Hz, respectively, whereas the vertical arrows indicate the artefacts due to switching on and off the square stimulus of 3 ms duration. (**b**) Average maximum AP upstroke velocity of five cardiomyocytes under control conditions and in the presence of 3 µM paroxetine (gray and black bars, respectively) at pacing frequencies of 1 and 3 Hz. * *p* < 0.05, ** *p* < 0.01, *** *p* < 0.001, two-way repeated measures ANOVA. (**c**) Percent decrease in maximum AP upstroke velocity at a pacing frequency of 3 Hz as compared to 1 Hz under control conditions and in the presence of 3 µM paroxetine (gray and black bars, respectively). ** *p* < 0.01, paired *t*-test.

**Figure 6 ijms-24-01904-f006:**
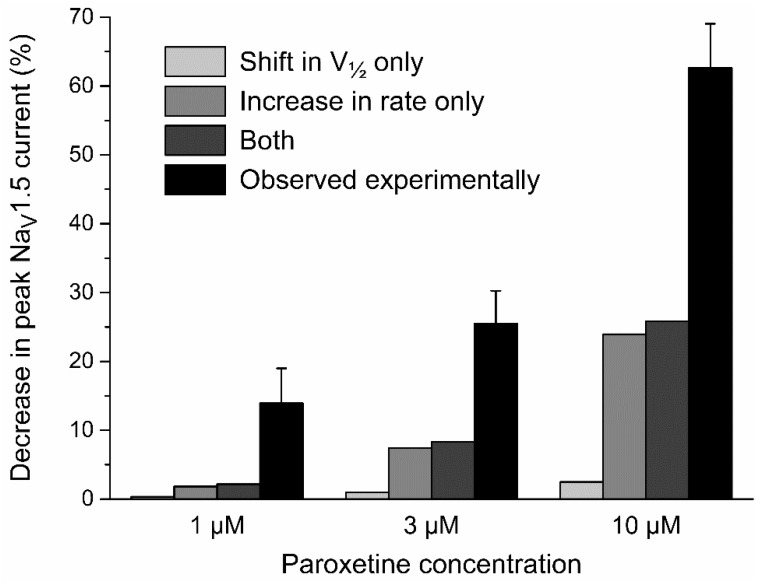
Paroxetine-induced decrease in peak Na_V_1.5 current observed experimentally (black bars) and in computer simulations of voltage clamp experiments, applying the voltage clamp protocol of Figure 1a to the fast sodium current of the human left ventricular cell model by Ten Tusscher et al. [42], as updated by Ten Tusscher and Panfilov [43] (gray bars). Simulations were carried out with shifts in the half-inactivation voltage V_½_ per se (−3.3, −7.3, and −12.4 mV, when simulating paroxetine concentrations of 1, 3, and 10 µM, respectively), increases in inactivation rate per se (6%, 27%, and 102%, when simulating paroxetine concentrations of 1, 3, and 10 µM, respectively), or both.

**Figure 7 ijms-24-01904-f007:**
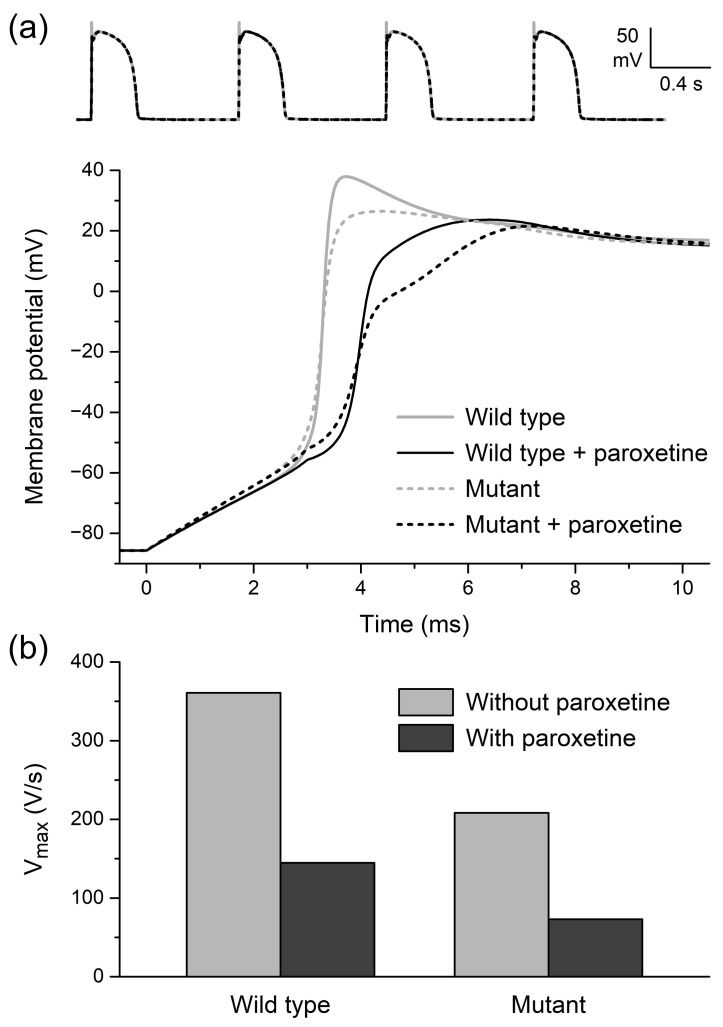
Effects of paroxetine on action potentials of single human left ventricular cardiomyocytes in computer simulations, using the human left ventricular cell model by Ten Tusscher et al. [42], as updated by Ten Tusscher and Panfilov [43]. (**a**) Action potentials elicited at 1 Hz (top) and associated upstroke phase (bottom) of wild-type and Na_V_1.5 mutant cardiomyocytes in the absence or presence of paroxetine. (**b**) Associated maximum upstroke velocity (V_max_).

**Figure 8 ijms-24-01904-f008:**
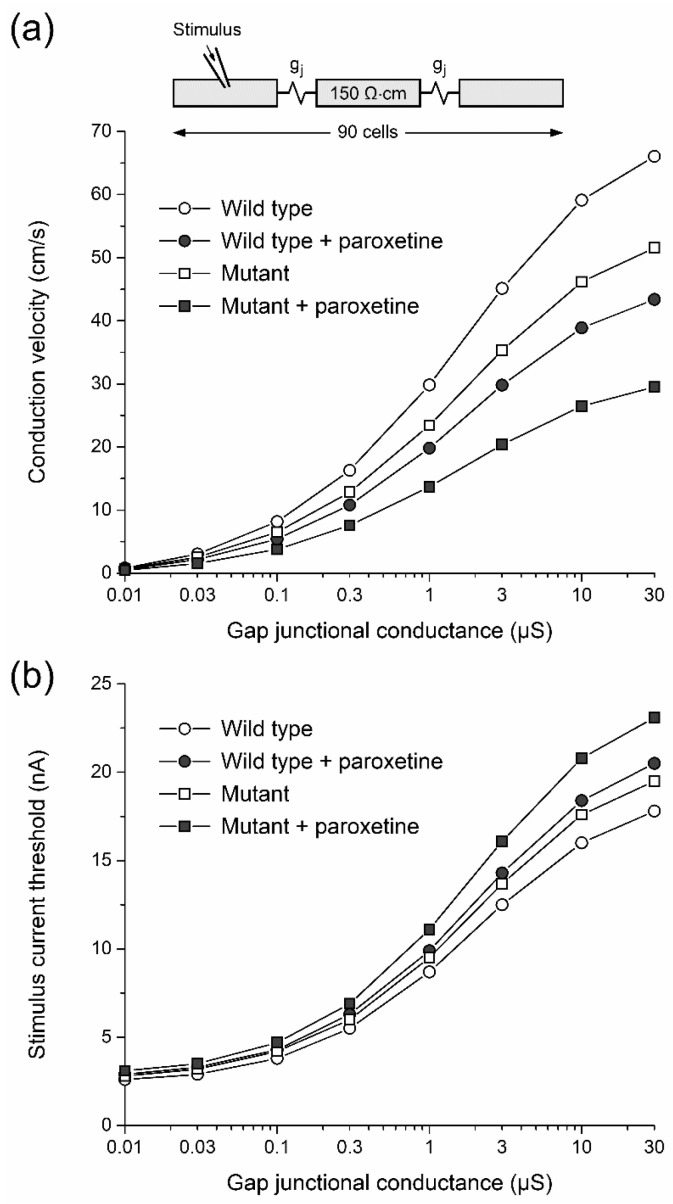
Effects of paroxetine on action potential conduction in computer simulated one-dimensional strands of human left ventricular cardiomyocytes. (**a**) Conduction velocity in strands of wild-type and Na_V_1.5 mutant cardiomyocytes in the absence or presence of paroxetine as a function of gap junctional conductance (gj). Individual cells of the 90 cell strand (inset) described according to the human left ventricular cell model by Ten Tusscher et al. [42], as updated by Ten Tusscher and Panfilov [43]. Myoplasmic resistivity set to 150 Ω∙cm [44]. (**b**) Associated stimulus current threshold for successful conduction. Stimulus current of 2 ms duration applied to the leftmost cell of the strand at a frequency of 1 Hz.

## Data Availability

Data will be available after publication upon request to academic researchers.

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
