# Peer review of "The Antidepressant Paroxetine Reduces the Cardiac Sodium Current"

_ijms, 2023, doi:10.3390/ijms24031904_

Round 1

Reviewer 1 Report

The manuscript by Plijter et al., entitled “The Antidepressant Paroxetine Reduces the Cardiac Sodium Current” examines the effect of the anti-depressant drug on cardiac sodium current.   Electrophysiological recordings were made on both rabbit ventricular myocytes and HEK293 cells stably expressing NaV1.5 channels.  Mathematical modelling was conducted using Ten Tusscher and Panfilov human ventricular cell model.  In HEK293 cells, paroxetine reduced NaV1.5 current in a dose-dependent fashion (IC50 ~ 6.9 µM), and shifted steady state inactivation in the hyperpolarizing direction.  In rabbit ventricular myocytes, paroxetine (3 µM) decreased Vmax and AP amplitude.  Computer simulations confirmed that paroxetine reduces the fast sodium current of human left ventricular cardiomyocytes, thereby slowing conduction and reducing excitability in strands of cells.  These effects are further exacerbated when a loss-of-function mutation in NaV1.5 in incorporated into the model.  The authors conclude that paroxetine acts as an inhibitor of NaV1.5 channels, which may be enhanced when a SCN5A loss-of-function mutation is present.

General Comments

I thought the manuscript was clearly written and experiments appear to be carefully performed.  I do have some suggestions for the authors to consider, some of which may require additional experiments. 

Major Comments.

1) I do have some concerns with the number of cells used in each phase of the study.  Ideally, the authors should report n values of at 7-8 cells.  I would suggest increase the replicates in some of the Figures (for example, Figures 1, 2 and 4).

2) Many sodium channels inhibitors shift steady-state inactivation to more negative potentials (as the authors nicely show in Figure 2) and slow recovery from inactivation (reference).  I noted the authors applied the voltage clamp protocols every 5 seconds.  Do the authors have any evidence that a 5 s interpulse interval is sufficient to achieve steady-state conditions?  A recovery from inactivation curve in the absence and presence of paroxetine would conclusively show this.    Although it seems unlikely that there is any use-dependent block with a 5 s interpulse interval and a holding potential of -120mV, this should be demonstrated.  At the very least, a statement should be made in the text.

Minor Comments

1)  I noted in the IV curve (Figure 1) that the reversal potential was around -5 mV.  Given that external Na+ is 20mM and internal Na is 14mM, this should result in a reversal ~+5mV.  Was offset potential accounted for and corrected?

2)  In the HEK293 cells, why was Na+ current recorded under low Na+ conditions?  Judging by the size of the representative traces depicted in Figure 1, voltage control should be achievable with full Na+ in the external solution.  Alternatively, since rabbit ventricular myocytes were isolated for this study, why not use the myocytes to record Na+ current under low external Na conditions.

3) The errors bars are quite large in the steady-state inactivation curve shown in Figure 2C.  This may be a figure in which the n values could be increased.

4)  Did the author test the effects of paroxetine at any other concentrations or additional pacing rates in rabbit ventricular myocytes?  Do the authors see a more pronounce effect of paroxetine at faster pacing rates.5

5) Figures.  Capitalized the first letter in the axis labels.  Also, please add the concentration of paroxetine in Figure 4a.

Author Response

We thank the reviewer for his/her time and efforts to review our manuscript and his/her constructive comments. We took the reviewer’s comments to heart and made changes to the manuscript accordingly. Our response to each of the reviewer’s specific comments is given below, repeating the reviewer’s comment in bold, followed by our response. Changes made to the manuscript are detailed here and appear in the revised manuscript as ‘tracked changes’ through the ‘Track Changes’ function of MS Word, as requested by the editors.

The manuscript by Plijter et al., entitled “The Antidepressant Paroxetine Reduces the Cardiac Sodium Current” examines the effect of the anti-depressant drug on cardiac sodium current.  Electrophysiological recordings were made on both rabbit ventricular myocytes and HEK293 cells stably expressing NaV1.5 channels.  Mathematical modelling was conducted using Ten Tusscher and Panfilov human ventricular cell model.  In HEK293 cells, paroxetine reduced NaV1.5 current in a dose-dependent fashion (IC50 ~ 6.9 µM), and shifted steady state inactivation in the hyperpolarizing direction.  In rabbit ventricular myocytes, paroxetine (3 µM) decreased Vmax and AP amplitude.  Computer simulations confirmed that paroxetine reduces the fast sodium current of human left ventricular cardiomyocytes, thereby slowing conduction and reducing excitability in strands of cells.  These effects are further exacerbated when a loss-of-function mutation in NaV1.5 in incorporated into the model.  The authors conclude that paroxetine acts as an inhibitor of NaV1.5 channels, which may be enhanced when a SCN5A loss-of-function mutation is present.

General Comments

I thought the manuscript was clearly written and experiments appear to be carefully performed.  I do have some suggestions for the authors to consider, some of which may require additional experiments. 

Major Comments.

1) I do have some concerns with the number of cells used in each phase of the study.  Ideally, the authors should report n values of at 7-8 cells.  I would suggest increase the replicates in some of the Figures (for example, Figures 1, 2 and 4).

The reviewer is right that the number of cells used in each phase of the study is fairly small. However, despite this apparent limitation, the effects of paroxetine on NaV1.5 density and gating properties were significant and consistently present at multiple drug concentrations. In addition, the AP measurements confirmed this principle finding and further demonstrated that it is also present in a cardiomyocyte background under close-to-physiological conditions. Because all experiments were carried out as paired ones, the power of statistics was considerably raised at our small numbers of experiments. We have now included a paragraph addressing this issue, and other potential limitations of our study, at the end of the Discussion section (page 17, lines 510­–522). Of note, the number of cardiomyocytes used for Figure 4 was increased from 4 to 6 by analyzing recordings from two additional myocytes. For 5 of these 6 cardiomyocytes, AP recordings were not only made at a stimulus frequency of 1 Hz, but also at 3 Hz. The latter recordings were now analyzed and the obtained results are presented in our new Figure 5.

2) Many sodium channels inhibitors shift steady-state inactivation to more negative potentials (as the authors nicely show in Figure 2) and slow recovery from inactivation (reference).  I noted the authors applied the voltage clamp protocols every 5 seconds.  Do the authors have any evidence that a 5 s interpulse interval is sufficient to achieve steady-state conditions?  A recovery from inactivation curve in the absence and presence of paroxetine would conclusively show this.  Although it seems unlikely that there is any use-dependent block with a 5 s interpulse interval and a holding potential of -120mV, this should be demonstrated.  At the very least, a statement should be made in the text.

We thank the reviewer for pointing to this issue. We have not performed experiments to measure the recovery from inactivation curve in the absence and presence of paroxetine. However, from previous experiments by Amin et al. [52] and ongoing measurements (not yet published), we know that the recovery from inactivation of NaV1.5 currents in HEK-293 cells at room temperature is complete within 1 second when using a holding potential of −120 mV. This makes it highly unlikely that changes in recovery from inactivation contribute to our observed decrease in NaV1.5 currents in HEK-293 cells. We have now included the following sentence on page 19 (lines 602­–604): “Cycle length was 5 s, which results in full recovery from inactivation at room temperature when using a holding potential of −120 mV [52].” However, under more physiological conditions, i.e., at a temperature of 37 °C and a holding potential of −85 mV, recovery from inactivation is much slower and not fully complete within 1 second (see Figure 4 of Berecki et al. [35]). Therefore, changes in recovery from inactivation may have contributed to the effects of paroxetine on the maximum upstroke velocity (Vmax) of the cardiomyocyte APs at 1 Hz. This seems substantiated by the significantly more pronounced reduction in Vmax that we observed at an AP stimulus frequency of 3 Hz. At 1 Hz and 3 Hz, the reduction in Vmax amounted to 16.0 ± 4.5% and 33.1 ± 2.6% (mean ± SEM, n = 5), respectively (p < 0.01, paired t-test). These additional data are included in our new Figure 5 and described in the Results section on page 10–11 (lines 288­–309). In addition, this finding is brought up in the Discussion section on page 16 (lines 444­–446).

 Minor Comments

1)  I noted in the IV curve (Figure 1) that the reversal potential was around -5 mV.  Given that external Na+ is 20mM and internal Na is 14mM, this should result in a reversal ~+5mV.  Was offset potential accounted for and corrected?

We thank the reviewer for this insightful comment. Although there is always some cell-to-cell variation in the measured NaV1.5 reversal potential, the reviewer is right that the measured reversal potential in Figure 1b is slightly more negative than the calculated reversal potential for Na+, based on our used solutions and temperature. We have not used liquid junction potential correction for the NaV1.5 current, but this cannot account for the difference. We have now reanalyzed our data to check for potential errors in the analysis, but it appeared that the data were correctly implemented in the original manuscript. In the present study, we used relatively small HEK-293 cells (with a membrane capacitance of 7.9 ± 0.7 pF (n = 11)) and we cannot exclude that the Na+ influx through the NaV1.5 channels, which are expressed in high density, gives higher local Na+ concentrations in such small cells. Further studies are required to resolve this issue. Nevertheless, the reversal potential in absence and presence of 3 µM paroxetine did not differ significantly (−2.0 ± 1.8 mV (control) vs. −4.6 ± 2.4 mV (3 µM paroxetine); p > 0.05, paired t-test). Thus, we exclude a reduction in driving force as mechanism for the smaller NaV1.5 currents in presence of paroxetine. This information is now included in subsection 2.1.1 of the Results section (page 4, lines 145­–148).

 2)  In the HEK293 cells, why was Na+ current recorded under low Na+ conditions?  Judging by the size of the representative traces depicted in Figure 1, voltage control should be achievable with full Na+ in the external solution.  Alternatively, since rabbit ventricular myocytes were isolated for this study, why not use the myocytes to record Na+ current under low external Na conditions.

In some previous studies, including those by Coronel et al. (2005, DOI: 10.1161/CIRCULATIONAHA.105.532614), Amin et al. [52], and Berecki et al. [35], we indeed used a close-to-physiological extracellular Na+ concentration, even at 37 °C, to measure NaV1.5 currents in HEK-293 cells. However, such NaV1.5 current measurements were performed in transiently transfected HEK-293 cells, which show much lower NaV1.5 current densities than HEK-293 cells with stable NaV1.5 expression. In previous (Portero et al. [63]; Man et al. (2021, DOI: 10.1161/CIRCULATIONAHA.121.054083)) as well as ongoing studies using HEK-293 cells with stable NaV1.5 expression, we therefore lowered the extracellular Na+ concentration to 20 mM to obtain reliable voltage clamp. We have now included this information at the start of subsection 4.2.2 of the Materials and Methods section (page 19, lines 594­–596), reading: “We selected relatively small HEK-293 cells (with a membrane capacitance of 7.9 ± 0.7 pF (n = 11)) and used a bath solution with a relatively low Na+ concentration to obtain reliable voltage clamp.”

Our experiments on freshly isolated rabbit cardiomyocytes were performed to test the effect of paroxetine on APs in a cardiomyocyte background under close-to-physiological conditions rather than to test the effects of the drug on all NaV1.5 current properties in detail. In the revised version of the manuscript, this information is now present at the start of subsection 2.2 of the Results section (page 9, lines 257–261).

 3) The errors bars are quite large in the steady-state inactivation curve shown in Figure 2C.  This may be a figure in which the n values could be increased.

The reviewer is right about the size of the error bars in Figure 2c. The relatively large error bars are due to cell-to-cell variability in steady-state inactivation, while our experimental design is such that we have tested the effects of paroxetine on steady-state inactivation in individual cells with the use of paired experiments, directly comparing paroxetine to control in each individual cell. Increasing the number of cells, as suggested by the reviewer, may decrease the absolute size of the error bars in Figure 2c, but will likely not affect the significance of the paroxetine-induced negative shift of the steady-state inactivation curve at 3 and 10 µM, which is already observed at n values of 5 and 3, respectively.

 4)  Did the author test the effects of paroxetine at any other concentrations or additional pacing rates in rabbit ventricular myocytes?  Do the authors see a more pronounce effect of paroxetine at faster pacing rates.

The reviewer addresses a relevant issue. As set out above in our response to ‘major point #1’, AP recordings were not only made at a stimulus frequency of 1 Hz but also at 3 Hz, and these recordings have now been analyzed. As shown in our new Figure 5, and already mentioned in our response to ‘major point #2’, the reduction in maximum upstroke velocity is more pronounced at a pacing frequency of 3 Hz than at 1 Hz. Because of the limited availability of rabbit ventricular myocytes, we could only test the effect of paroxetine at a single concentration.

 5) Figures.  Capitalized the first letter in the axis labels.  Also, please add the concentration of paroxetine in Figure 4a.

We have now capitalized the first letter in each of the axis labels and we have also included the concentration of paroxetine in Figure 4a (as well as in our new Figure 5).

Reviewer 2 Report

Major

1, the introduction is extremely long, almost like a review paper. I would recommend a little bit of focusing and moving some of the content  into the discussion

2, current was reduced over the entire voltage range in Fig 1b. Looking at the figure it does not seem to be the case.

3, The dose response in Fig. 1 is  for 3 data points. This is quite unusual and difficult to accept considering the number of free parameters used in the fitting. The number of free parameters for the fit may exceed the number of data points, although the actual equation used for  fitting is not shown- I recommend showing the equation and the number of free parameters.

Is there any reason for using 3 concentration only? One must use pharmacological dilution series, and at least 5 points for a dose-response.

4, If IC50 is almost 7 uM.  Why did the authors use 3 uM in the rest of the study? Please explain it in the manuscript.

5, For the voltage dependence of steady-state in activation, the second pulse (P2)was to -20 mV, whereas the peak sodium current is at around -45 mV (Fig.1). Why did the authors choose -20 mV for the read out of the remaining current rather than -45 mV where the current would be larger and more accurately determined?

6, The statistical analysis section describes that mean +/- SEM is displayed. The SEM seems to be very large for Fig. 2C, the error bars are overlapping. On the contrary, the statistical analysis reports significant differences, with p<0.001. Please indicate the statistical comparison method for this and all the other figures in the legend so that one can interpret the data without reading all the corresponding text and methods. Moreover, the list of statistical test in the methods section does not allow to isolate the statistical procedure used for a given decision.

7, The action potential duration seems to be shorter in the presence of the drug  based on the graph in Fig. 4a. Is it possible to put a record into the manuscript that would reflect the lack of effect of paroxetine on the APD? The current figure is inconsistent with the statistical analysis.

Minor:

Please rephrase this sentence, it is not clear what it means

„Even within a class of drugs developed to treat a specific non-cardiac disease, differences in risk of out-of-hospital cardiac arrest (OHCA) may exist. „

Please rephrase this sentence, to indicate that this shit is at 3 uM concentration.

The shift amounted to −7.3 ± 1.5 mV (n = 5).

Please rephrase the end of this sentence:

As illustrated in Figure 3a, we observed that the NaV1.5 current activation was not substantially affected by paroxetine, but that its inactivation was fastened.

Is this data shown somewhere in the manuscript or data not shown?

Repeating our analysis with voltage clamp steps to a test potential of −50 mV instead of −20 mV revealed highly similar changes in the rate of inactivation in response to paroxetine.

Author Response

We thank the reviewer for his/her time and efforts to review our manuscript and his/her constructive comments. We took the reviewer’s comments to heart and made changes to the manuscript accordingly. Our response to each of the reviewer’s specific comments is given below, repeating the reviewer’s comment in bold, followed by our response. Changes made to the manuscript are detailed here and appear in the revised manuscript as ‘tracked changes’ through the ‘Track Changes’ function of MS Word, as requested by the editors.

Major

 1, the introduction is extremely long, almost like a review paper. I would recommend a little bit of focusing and moving some of the content  into the discussion.

We thank the reviewer for her/his suggestion. We tried to follow the reviewer’s suggestion, but regretfully found that we could only remove the last paragraph of the introduction, which reduced its length somewhat. We found the remaining paragraphs of the Introduction section all relevant for our paper that we submitted for publication in the special topic “State-of-the-Art Molecular Pharmacology in Netherlands”. Likely, this particular special topic will attract readers with a diverse background, making it essential to explain the backgrounds in some detail.

 2, current was reduced over the entire voltage range in Fig 1b. Looking at the figure it does not seem to be the case.

We thank the reviewer for pointing to our incorrect wording in the description of Figure 1b in subsection 2.1.1 of the Results section. Indeed, paroxetine significantly reduced NaV1.5 currents only when a substantial NaV1.5 current was present, such as in the −60 to −15 mV voltage range. We have adapted the particular sentence, which now reads (page 4, lines 140–142): “The peak NaV1.5 current reached its maximum amplitude near −45 mV and was significantly decreased by paroxetine over a wide voltage range (…).”

 3, The dose response in Fig. 1 is  for 3 data points. This is quite unusual and difficult to accept considering the number of free parameters used in the fitting. The number of free parameters for the fit may exceed the number of data points, although the actual equation used for  fitting is not shown- I recommend showing the equation and the number of free parameters.

Is there any reason for using 3 concentration only? One must use pharmacological dilution series, and at least 5 points for a dose-response.

We apologize for being unclear. Although the fitting equation was already included in subsection 4.2.2 of the Materials and Methods section, we have now rephrased the last few sentences of subsection 2.1.1 of the Results section to explain our fitting procedure into more detail (page 4, lines 153–159). Please note that we have normalized our data to the amount of block, which is in the range between 0 (baseline conditions) and 100% (full blockade), and have data points at three concentrations on the slope of the dose-response curve. According to Turner & Charlton [62], such a small amount of concentrations is sufficient for standard sigmoidal dose-response curves. In addition, our used concentrations are around the IC50, which enhances the reliability of the IC50 determination. Nevertheless, we agree that in most studies a higher amount of concentrations is used, especially with actually measured data points at the plateau levels at either end of the curve.  We have now included a paragraph addressing this issue, and other potential limitations of our study, at the end of the Discussion section (page 17, lines 510–522).

 4, If IC50 is almost 7 uM.  Why did the authors use 3 uM in the rest of the study? Please explain it in the manuscript.

All experiments on cardiomyocytes were completed before we were able to determine the IC50 in detail, which made the use of a concentration closer to the IC50 practically impossible. In Figures 2b,d and 3b–d, we summarize NaV1.5 current parameters at all three concentrations used. For the AP measurements, we could only test the effect of paroxetine at a single concentration because of the limited availability of rabbit ventricular myocytes. We chose a concentration of 3 µM because at this concentration we had already observed a considerable amount of NaV1.5 current changes in our initial series of voltage clamp experiments on HEK-293 cells without too extreme effects that might prevent excitability of cardiomyocytes completely. The latter information is now included at the start of subsection 2.2 of the Results section (page 9, lines 261–264).  

 5, For the voltage dependence of steady-state in activation, the second pulse (P2)was to -20 mV, whereas the peak sodium current is at around -45 mV (Fig.1). Why did the authors choose -20 mV for the read out of the remaining current rather than -45 mV where the current would be larger and more accurately determined?

For the determination of steady-state inactivation of sodium current, it is essential to have a P2 step to a potential were all sodium channels are activated under baseline as well as during interventions, such as drug application. At −45 mV, the NaV1.5 current amplitude is indeed larger (due to a larger driving force), but at this potential not all channels are yet activated, as can be seen in Figure 2a. Therefore, a potential of around −20 mV is commonly used in NaV1.5 patch-clamp studies, including ours.

 6, The statistical analysis section describes that mean +/- SEM is displayed. The SEM seems to be very large for Fig. 2C, the error bars are overlapping. On the contrary, the statistical analysis reports significant differences, with p<0.001. Please indicate the statistical comparison method for this and all the other figures in the legend so that one can interpret the data without reading all the corresponding text and methods. Moreover, the list of statistical test in the methods section does not allow to isolate the statistical procedure used for a given decision.

The reviewer is right about the size of the error bars in Figure 2c. The relatively large error bars are due to cell-to-cell variability in steady-state inactivation, while our experimental design is such that we have tested the effects of paroxetine on steady-state inactivation in individual cells with the use of paired experiments, directly comparing paroxetine to control in each individual cell. We have followed the reviewer’s suggestion and have now specified the statistical tests used in the figure legends to clarify our statistical testing in more detail.

 7, The action potential duration seems to be shorter in the presence of the drug  based on the graph in Fig. 4a. Is it possible to put a record into the manuscript that would reflect the lack of effect of paroxetine on the APD? The current figure is inconsistent with the statistical analysis.

We followed the reviewer’s suggestion and have now selected another AP, which better reflects the lack of effect of paroxetine on the APD, to appear in Figure 4a.

 Minor:

 Please rephrase this sentence, it is not clear what it means „Even within a class of drugs developed to treat a specific non-cardiac disease, differences in risk of out-of-hospital cardiac arrest (OHCA) may exist.„

We apologize for being unclear and have modified the sentence into: “Even among commonly used drugs to treat a specific non-cardiac disease, differences in risk of out-of-hospital cardiac arrest (OHCA) may exist.”

Please rephrase this sentence, to indicate that this shit is at 3 uM concentration. The shift amounted to −7.3 ± 1.5 mV (n = 5).

We agree and have modified this sentence, which now reads: “The shift at a paroxetine concentration of 3 μM amounted to −7.3 ± 1.5 mV (n = 5).”

 Please rephrase the end of this sentence: As illustrated in Figure 3a, we observed that the NaV1.5 current activation was not substantially affected by paroxetine, but that its inactivation was fastened.

We followed the reviewer’s suggestion and modified this specific sentence, which now reads: “As illustrated in Figure 3a, we observed that the activation of the NaV1.5 current was not substantially affected by 3 µM paroxetine. However, its rate of inactivation was higher in the presence of the drug, resulting in NaV1.5 currents of shorter duration.”

 Is this data shown somewhere in the manuscript or data not shown? Repeating our analysis with voltage clamp steps to a test potential of −50 mV instead of −20 mV revealed highly similar changes in the rate of inactivation in response to paroxetine.

These data are indeed not shown in the manuscript. We had left a “data not shown” statement out because we had experienced, at the page proof state, that the editors of the Int. J. Mol. Sci. journal do not allow such statement. However, we agree with the reviewer that such statement can be helpful to the reader and have added it to the revised manuscript.

Round 2

Reviewer 1 Report

The revised manuscript by Plijter et al., entitled “The Antidepressant Paroxetine Reduces the Cardiac Sodium Current” examines the effect of the anti-depressant drug on cardiac sodium current.   Electrophysiological recordings were made on both rabbit ventricular myocytes and HEK293 cells stably expressing NaV1.5 channels.  Mathematical modelling was conducted using Ten Tusscher and Panfilov human ventricular cell model. 

General Comments

My previous comments have been adequately addressed.  No new comments in the revision.

Reviewer 2 Report

The authors have addressed the critiques, I still would have been more happy to see a full dose response curve but the authors argued against it which I  accept. In my opinion, doing an easy experiment is always better than arguments.